# Solid Concentration Estimation by Kalman Filter [note 1]

**DOI:** 10.3390/s20092657

**Published:** 2020-05-06

**Authors:** Yongguang Tan, Shihong Yue

**Affiliations:** School of Electrical and Information Engineering, Tianjin University, Tianjin 300072, China; tyg_1991@tju.edu.cn

**Keywords:** solid concentration, solid–liquid two-phase flow, Kalman filter, data fusion

## Abstract

One of the major tasks in process industry is solid concentration (SC) estimation in solid–liquid two-phase flow in any pipeline. The *γ*-ray sensor provides the most used and direct measurement to SC, but it may be inaccurate due to very local measurements and inaccurate density baseline. Alternatively, under various conditions there are a tremendous amount of indirect measurements from other sensors that can be used to adjust the accuracy of SC estimation. Consequently, there is complementarity between them, and integrating direct and indirect measurements is helpful to improve the accuracy of SC estimation. In this paper, after recovering the interrelation of these measurements, we proposed a new SC estimation method according to Kalman filter fusion. Focusing on dredging engineering fields, SCs of representative flow pattern were tested. The results show that our proposed methods outperform the fused two types of measurements in real solid–liquid two-phase flow conditions. Additionally, the proposed method has potential to be applied to other fields as well as dredging engineering.

## 1. Introduction

Solid concentration (SC) estimation plays an important role in the detection of pipeline multiphase flow [1,2], essentially in dredging engineering [3]. The high precision of SC is necessarily required to effectively control industrial processes, while a small error of estimated SC can cause huge economic cost and efficiency repercussions.

The SC estimation method has evolved for many years, and each progress provides information of better accuracy and robustness [4]. In dredging engineering, the *γ*-ray sensor is the most used meter to directly measure SC values. When solid and liquid phases are uniformly distributed in the pipe, the SC estimation is effective according to the measuring principle of the *γ*-ray sensor. However, in most cases, the solid–liquid mixtures are unevenly distributed in pipelines, and thus the measured SC values may be very inaccurate [5]. Additionally, the necessary density baseline regarding zero value of SC for any sensor is difficult to be effectively determined [6]. These problems are impossible to be solved by the technique of the *γ*-ray sensor, and may cause great errors to mislead further control processes.

In order to overcome the above problems, the indirect measurements from other sensors must be used. There are many measurements that can be used to adjust the accuracy of SC estimation. According to determinable interrelations to SC estimation, these measurements consist of a set of prior information. In view of some commonly approximating methods, such as least squares estimation method [7], linear minimum variance estimation (LMVE) method [8], maximum likelihood estimation method [9], and so on, some regression equations have been set up. When the *γ*-ray sensor is severely inaccurate, these equations may have higher accuracy than the direct SC measurements. Moreover, the equation can provide a stable estimation in most cases. However, these existing regression equations are limited by some uncertain assumptions and conditions. For example, LMVE can only access linear rather than nonlinear relations [8].

Both direct and indirect measurements have their own applicable ranges, and there is no clear boundary between them. Nevertheless, there is complementarity between them, and integrating the two types of measurements is helpful to improve the accuracy of SC estimation. Along this direction, the Kalman filter (KF) prediction [10] is the most used information fusion method, since it is a recursive, self-regression, optimal process in the industrial field [11]. The use of KF for information fusion in multi-phase flow is not new. Tan et al. [12] used a conductance ring coupled cone meter as a measuring sensor in an estimation model and thus improved the estimation accuracy of gas concentration in the gas–liquid. Wu et al. [13] developed an online adaptive estimation method for oil–water two-phase flow water holdup measurement with a conductance/ capacitance sensor. However, these methods hardly involve the characteristics of the used sensor, and specially are irrelative to the *γ*-ray sensor. Essentially, the SC estimation in dredging engineering fields is more complicated and difficult than these SC estimations in other conditions and flow patterns.

In this paper, we recover the interrelations of other measurements regarding SC estimation. Different from the existing methods, SC is computed not only by direct measurements but also by indirect estimations. We propose a new SC estimation method according to KF fusion without limited assumption and unreasonable conditions, and the complementarity between the two types of measurements are analyzed. Focusing on dredging engineering, key principles and evaluation criteria are introduced, and whereby typical SCs of the typical flow patterns were tested. Hence, the proposed method uses the complementarity between direct measurement and indirect measurements to improve measurement accuracy of SC estimation. The proposed method aims to obtain better accuracy to the SC estimation in dredging engineering.

## 2. Related Work

In this section, we introduce direct measuring principle of SC from the *γ*-ray sensor, indirect measurements from other sensors, and the KF approach below.

### 2.1. Direct SC Measurement from the γ-ray Sensor

The principle of a *γ*-ray source sensor is to install two ends of a ray source on a silt conveying pipe, one section transmits a signal, and one end receives a signal. The measurements of the *γ*-ray source changes when the sediment passes through the transmission pipe, and the key parameters such as the density and flow of the sediment are derived by the change of the signal.

The principle of the radiation attenuation is the first foundation of the *γ*-ray sensor. Assuming that the *γ*-rays are monoenergetic from a parallel beam (see Figure 1), the residual intensity of *γ*-rays is attenuated by the materials that the *γ*-rays pass through:
(1)$$I=I_{0}\exp\{-\int_{L}u_{m}\rho ds\},$$
where *u_m_* the mass attenuation coefficient (m^2^/kg) mainly due to the photoelectric, pair production effects and so on; *ρ* is the density of the object (kg/m^3^); *ds* represents the infinitesimal thin layer of the object (m).

When Equation (1) is used for SC estimation, *I* is available measurement from the *γ*-ray sensor; *I*_0_ is the baseline value that refers to zero SC value. Consequently, the measuring intensity of the *γ*-ray sensor is directly proportional to the density of all materials in the field that the *γ*-ray sensor goes through. Assume that all materials in the entire field are uniformly distributed, then the investigated intensity can be regarded as the density of the global field when *I*_0_ is converted to zero density of SC. The SC value can be solved by Equation (1).

However, the CS estimation based on the *γ*-ray sensor may be inaccurate due to the following two reasons at least.
(1)Uneven distribution. Both solid and liquids are assumed to be uniformly distributed in the global field, and thus the local measurement that *γ*-rays go through is regarded as the global one (see Figure 1). However, in most cases the local density that the *γ*-ray sensor measures are very different from the global density for asymmetric and non-uniform flows.(2)Uncertain baseline. The estimated SC greatly depends on the comparable baseline that responds to the zero SC value. However, the density baseline is difficult to determine in advance. For example, the *γ*-ray intensity in seawater is taken as the value of *I*_0_, but the seawater density is changeable in a large range from 1.02 to 1.08 g/cm^3^ in various conditions. Since seawater occupies more than half of space in the pipe, thus, its density may lead to large errors of the SC estimation.

### 2.2. Indirect SC Estimation from Other Sensors

Due to the limitations of SC measurement *ξ* by the *γ*-ray sensor, indirect measurements from other sensors must be used to adjust the SC value. In a dredging engineering, there empirically are relative measurements from 13 other sensors, *Q_i_, I* = 1, 2, …, 13, as shown in Table 1.

In order to effectively assess the correlations from 13 indirect measurements to SC, we calculated their correlation coefficients that are formulated as
(2)$$\rho (\xi ,Q_{k})=\mathrm{cov}(\xi ,Q_{k})/(\sqrt{D(\xi })\sqrt{D(Q_{k}})),\;k=1,\;2,\;\ldots ,\;13$$
where cov(·) is covariance, and *D*(·) is variance on *ξ* and *Q_k_*, respectively. The third column in Table 1 shows computed correction coefficients.

Specially, Figure 2 shows the measured values of *ξ* and *Q*_10_. It can be observed that the varying trend of *Q*_10_ can access one of *ξ* in a very wide range. In the next section, we will illustrate that if all measurements of *ξ* are smoothed, *Q*_10_ can play an important role in the SC estimation in dredging engineering [14].

According to these determinable correlations to SC and commonly approximating methods, some equations have been set up, which can be formulated as
(3)$$\xi =F(Q_{1},\;Q_{2},\;\ldots ,\;Q_{13})$$
where *F* denotes a map from measurements to SC value *ξ*. However, the correctness of Equation (3) may be affected by measuring conditions and construction form of *F*. When measuring conditions and construction form are changed, these equations may cause a large error. An effective method is to directly establish the correlation from these measurements to SC [15].

Direct and indirect measurements have interrelations as below:(1)When the two measurements are nearly consistent, their results are believable.(2)If the two measurements are inconsistent, they have individual effective ranges and complementarity under various conditions. Generally, when any measurement has very large or sudden change, it is unbelievable [16]. In fact, the solid–liquid mixtures have been sieved by an iron net and broken by water cannons before they are transported into the pipe. Therefore, the density change in the pipe is relatively stable. Hereafter, we apply the complementarity to the KF estimation in this paper.

### 2.3. KF Prediction

KF prediction is a linear estimation algorithm. In most cases, KF has demonstrated to be a recursive, self-regression, precise model in industrial fields.

Together with the controlling effect, KF results from two basic recursive equations from (*k* − 1) to *k* steps:
(4)$$X_{k}=A_{k}X_{k-1}+B_{k}U_{k}+V_{k}\;(state\;equation)$$
and
(5)$$Z_{k}=H_{k}Z_{k-1}+W_{k}\;(observation\;equation)$$
where *X_k_* is the state vector that is governed by state matrix *A_k_* of the system, *Z_k_* is the observation (measurement) vector that satisfies observation matrix *H_k_*, *V_k_* and *W_k_* are system noise and observation noise vectors, respectively; *B_k_U_k_* is a control item by coefficient vector *U_k_*; both *V_k_* and *W_k_* hypothetically are white noise with zero mean and individual covariance in the case of no prior information.

At the time *k,* assume that ${\hat{X}}_{k}$ is the optimal estimation of *X_k_*, and ${\hat{P}}_{k}$ is the estimation of *P_k_*, then KF obeys the following five basic equations:
(6)$${\hat{X}}_{k}=A_{k}{\hat{X}}_{k-1}+B_{k}U_{k}+V_{k}\;(\mathrm{state}\;\mathrm{estimate})$$
(7)$${\hat{P}}_{k}=A_{k}{\hat{P}}_{k-1}A_{k}^{T}+Q_{k}\;(\mathrm{prediction}\;\mathrm{covariance})$$
(8)$$K_{k}={\hat{P}}_{k}H_{k}^{T}(H_{k}{\hat{P}}_{k}H_{k}^{T}+R_{k})\;(\mathrm{filter}\;\mathrm{gain})$$
(9)$$X_{k}={\hat{X}}_{k}+K_{k}(Z_{k}-H_{k}{\hat{X}}_{k})\;(\mathrm{state}\;\mathrm{update})$$
(10)$$P_{k}=(I-K_{k}H_{k}){\hat{P}}_{k-1}\;(\mathrm{covariance}\;\mathrm{matrix})$$


According to the above five equations, Table 2 shows the KF prediction process.

The KF prediction involves two alternative processes: state vector *X_k_* update and observation vector *Z_k_* update after recursively computing *P_k_*. KF has been demonstrated to be convergent without any other assumptions [11].

However, the existing methods that apply KF to the fusion process in multi-phase flow do not use the controlling effect from *U_k_*. Essentially, the two values in *V_k_* and *W_k_* are set to unchangeable constant. Hence, the applicable range of the used KF is very limited. In this paper, effort is made in dredging engineering, and thus both *X_k_* and *Z_k_* that act for the estimation of SC are only scalar data, denoting them as *x_k_* and *z_k_*. To effectively apply KF for the SC estimation, two changes were made.
(1)The use of the controlling effect is to represent indirect measurements when the value of *A_k_* is used to reflect inherent characteristics hidden in various measurements, and simultaneously can show their dynamic changes.(2)Self-adjustable covariance in *X_k_* and *Z_k_* in KF are used to show trusting degrees from direct and indirect measurements in the *γ*-ray sensor and in other sensors, respectively. Most existing KF approaches apply the unchangeable values of covariance, and whereby the efficiency may greatly be decreased [17].

These changes in KF are explained in detail below.

## 3. The Proposed Method

In this section, we firstly illustrate how indirect measurements are used to estimate the value of SC in KF, and then explain how direct and indirect measurements can be fused toward more accurate SC estimation.

The SC estimation in dredging engineering obeys the following two principles:
(1)The use of SC aims to guide the control action online such that the amount of solid phase can reach to its largest value. However, since these control actions are operated by operators, they prefer to find varying trends of direct and indirect measurements rather than these values themselves.(2)The control action of *u_k_* is tightly close to all direct and indirect measurements. Consequently, iterative process of *x_k_* is greatly affected by the control action whose effect is to simulate the adjustment process of direct measurement by the 13 indirect measurements. Moreover, these measurements must be normalized and smoothed to recover natural characteristics of these measurements.

According to the above principle, after normalizing all measurements, the *i*-th SC value *ξ_i_* is smoothed by its five neighbors along the time order as
(11)$$\xi_{i}\leftarrow (\xi_{i-2}+\xi_{i-1}+\xi_{i}+\xi_{i+1}+\xi_{i+2})$$


Figure 3 shows a group of initial SC measurements with fast change become flat after smoothing them by Equation (11), and thereby the varying trend of SC can become clearer.

Specially, the correlations of 13 indirect measurements to SC can be greatly changed after smoothing. Figure 4 shows the change of correlations after smoothing them from one to three times.

Figure 4 shows that *Q*_1_, *Q*_2_, *Q*_3_, *Q*_4_, and *Q*_10_ are most relative to SC, and in fact they are independent to each other [18]. Each of them can be used to evaluate the working state of the *γ*-ray sensor in dredging engineering. Their correlation after smoothing three times is clearest, and thus these corrections are applied in our proposed approach.

In our proposed method, the KF approach uses Equation (4) to integrate other measurements to the SC estimation, and the KF approach thus is formulated as
(12)$$\xi_{k}=a_{k}\xi_{k-1}+b_{k}u_{k}+v_{k}$$
where *ξ_k_* is the estimated SC, *b_k_* is a coefficient that decides the effect of *u_k_* and can affect the converging speed of KF. Generally, both *a_k_* and *b_k_* must satisfy the following relation to assure the convergence of KF,
(13)$$(b_{k}+\varepsilon )/(a_{k}+\varepsilon )<1$$
where *ε* is a small positive number to perfect the rate from fast change [17]. If Equation (13) is not satisfied, then the value in *P_k_* in KF must be multiplied by a coefficient *g_k_*, i.e.,
(14)$$g_{k}=\max\{1,(c_{0}-u_{k}-v_{k})/v_{k}P_{k}w_{k}\}$$
where *c*_o_ is a positive number that can decide the converging speed of the KF approach and convergence. Consequently, in the proposed KF approach, there are four key parameters *a_k_, v_k_, w_k_*, and *u_k_*, and they are determined as follows.
(1)Determination of *a_k_*. The value of *a_k_* is tightly relative to the predicting mechanism of KF, and is used not only to predict *x_k_* but also to compute the error covariance *P_k_* in KF. Therefore, *a_k_* is the key parameter that reflects the varying trend of *x_k_* as the iteration proceeds.In the sense of probability, the varying trend of *x_k_* is consistent with the trend of two latest values *x*_*k*−1_ and *x*_*k*−2_. The value of *a_k_* is computed as
*a_k_* = *x_k_*_−1_/*x_k_*_−2_(15)
When *a_k_* is larger than 1, the value of *x_k_* is enlarged which can just show an increasing trend; otherwise, the value of *x_k+_*_1_ is reduced and corresponds to a deceasing trend.Figure 5 shows the effect of *a_k_* in a group of SC values, it can be observed that the values of *a_k_* can effectively represent the ascending and descending trends.(2)Determination of *u_k_*. The use of *u_k_* in KF aims to make the direct measurement of CF from the *γ*-ray sensor more accurate. Thus, *u_k_* is determined by prior information from other measurements.Let *Q*(*i*) = (*q*_1_(*i*), *q*_2_(*i*), …, *q*_13_(*i*))*^T^* be the *i*th vector on a group of measured values from *Q_i_*, *i* = 1, 2, …, 13; *w_i_* be the weighting value of *q_i_* by normalizing their correlations to SC in Table 1, respectively; ${\hat{s}}_{i}$ be the posteriori and more accurate real SC value relative to *Q*(*i*) after computing the total production by measuring ξ(i) at each working phase.Denoting (*Q*(*i*), ${\hat{s}}_{i}$) as the *i*th pair of the existing record, there is a tremendous amount of historical records whose set is denoted as *QS*. Consequently, we realize the relation *F* in Equation (3) by the basic principle “similar problems have similar solutions” in case-based reasoning model [19,20], and naturally it is a weighting average of all inputs.For any new indirect measurement *Q*(*i*), we can estimate its unknown value of SC value $s_{i}$ by a simple weighting mean as
(16)$$s_{i}=\sum_{j=1}^{m}d_{j}{\hat{s}}_{j},\;s.t,\;\sum_{j=1}^{m}d_{j}=1$$
where *d_j_* is computed by the similarity from *Q_i_* to *m* historical records that are the closest to *Q_i_*, and thus their similarity is measured by the following weighting distance:
(17)$$d_{j}={\{\sum_{k=1}^{13}w_{k}(q_{k}(j)-q(k))\}}^{1/2}$$
where *w_k_* is the weighting value of *Q_k_*, *k* = 1, 2, …, 13.Consequently, the computed SC by Equation (17) is taken as the value of *u_k_* in Equation (11).(3)Determination of *v_k_* and *w_k_*. Both *v_k_* and *w_k_* in KF are usually difficult to directly predict since we typically do not have the ability to directly observe the process that we are estimating. Then, without other prior information their values both are unchangeable in KF. In such case, both the estimation error covariance *P_k_* and the KF gain *K_k_* will stabilize quickly [17]. However, an unchangeable constant is unhelpful to quickly improve the accuracy of SC. Generally, effective KF performance can be obtained by tuning the value of *v_k_* and *w_k_* at time *k*. Notice that both direct and indirect measurements of SC are inaccurate when their values contain sudden change. Essentially, the measurement from the *γ*-ray sensor is inaccurate if the distributions of liquid and solids are uneven and thus is quickly changeable. According to the principle, we can determine the values of *v_k_* and *w_k_* for each time by their coefficients of autocorrelation from two latest and adjacent measurements, i.e.,
(18)$$v_{k}\propto E(X_{k}\cdot X_{k-1})\;\mathrm{and}\;w_{k}\propto E(Z_{k}\cdot Z_{k-1})$$
The smaller the $v_{k}$ or $w_{k}$ values, the higher the trust in the prediction and observation models. If their values are 0, it means that only the prediction or the observation model is trusted. Additionally, as their values decrease, KF will converge more easily, but when *v_k_* is reduced to a certain extent, continuing to decrease may cause the system to start diverging. Figure 6 shows the flowchart of the KF approach to the SC estimation.


## 4. Experiment

Experiments were performed in dredging process in a cutter suction dredger named as TIANJI. Figure 7 shows the *γ*-ray sensor which was used to provide direct measurement *ξ* of SC. It is installed on a horizontal transport pipeline with diameter 750 mm and sampling rate 150/s. The solids and liquids are mixed in the mixing pipe with adjustable seawater flow concentration; the inlet liquid velocity is 3–12 ms^−1^, and will be increased to 8–25 ms^−1^ as the flow develops. Additionally, the 13 indirect measurements (see Table 1) can provide an indirect estimation *s_i_* of SC by Equation (16) in real time, *I* = 1, 2, …,13, which can monitor the running state and provide the SC estimation as well.

In the experiment, a total of 55,450 historical records that consist of indirect measurements, and their corresponding values of SC were corrected by posteriori information. All measurements are normalized, but there are invalid data in the original records which must be removed. These records to be eliminated include the following three classes:
(1)Zero SC value. A zero SC value results from the regular phase at each hour in which the pipeline is pushed by pure seawater to prevent the transport pipe becoming clogged.(2)Inconsistent measurement. There are such cases that two very different indirect measurements respond to approximately the same SC value. Usually, these measurements may result from abnormal running state of some sensors, or are collected under a low signal-to-noise situation. Consequently, they cannot reflect natural characteristics in practice.(3)Abnormal SC value. When some high-density solids go through the cross-section of the *γ*-ray sensor, the SC value may expose sudden changes and lead to invalid measurements. The estimated density is inaccurate.

Finally, 43,350 valid records were kept in experiments, and these records were used to compute SC by Equation (16) when new indirect measurements were available.

### 4.1. KF Fusion

The *γ*-ray sensor collects direct measurements for SC online, while other 13 relative sensors can provide indirect measurements to estimate SC by Equation (16).

In the KF approach, the initial values of *x*_0_ are computed by their mean of all SCs in 43,350 valid records, respectively. Additionally, thereby *x*_0_ is taken as 1.66 ton·m^−3^, *v_k_* and *h_k_* are individually set to 10 and 1. The initial value of *p_k_* changes from 0.01 to 0.2.

To speed up KF, the initial value of *m* in Equation (16) must be set well. Note that the consistency between direct and indirect measurements on the SC estimation reflects the accuracy, and their rate ranges from 0.8 to 1.6. We let *m* = 3 and 7 which is directly proportional to the value of the rate.

According the dredging operating rules and the KF principle, the correctness of the estimated SC can be assessed by the following two conditions:
(1)Consistency test. When both direct and indirect measurements are consistent, the iterative variance in KF should be small enough compared with those values under the inconsistent case. The smaller the variance value, the higher the trust in the KF approach. Essentially, if variance of *v_k_* is 0, it means that only the direct measurement is trusted.(2)Amount constraint. The SC estimation finally aims to measure the total amount of solid mixtures after transporting the solid–liquid mixtures to specified positions. Therefore, the total amount in a determinable time can act for an objective index to evaluate the SC estimation.

Figure 8 shows the performance of the KF approach in 450 sets of real-time measurements. It is observed that the fused SC value *x_k_* by the KF approach is flatter than the two individual fused measurements. Specially, these original measurements contain more unreasonable values of sudden change no matter which the direct and indirect measurements are taken.

On the other hand, the volume amount of accumulating solid (soil) mixtures on each day is measurable and recorded in any dredging engineering. Generally, the interrelation between the value *ξ* of SC and the volume amount *T* obeys the following form,
(19)$$T=k\sum_{t=1}^{T}\xi_{t}\Delta V_{t}$$
where *k* is a conversion coefficient from density to volume, commonly taking it as 0.64, *ξ_t_* is the density value at *t*th sampling, and *T* is the total number of samples on one day, ∆*V_t_* is the volume at each sampling interval. Table 3 shows the collected historical volume amounts of four continuous days that correspond to the used data in experiments, respectively. Here, the value of SC is taken as the fused value *x_k_* in KF, the direct measurement is *ξ_k_* and the indirect measurement is *s_k_*, respectively.

In total, the computed amounts from *x_k_* are less than the real value, while the products from *ξ_k_* are larger. In contrast to their computed products by Equation (20), the fused value in KF is most close to the real amounts except the second day in which they are nearly consistent. The results show the KF estimation is more correct and valid.

### 4.2. Evaluation by the Relative Value of Q_10_

The value of *Q*_10_ refers to the suction vacuum value of the underwater pump and plays an important role in the controlling process in dredging engineering. Although the value of *Q*_10_ itself is not most directly relevant to the value of SC, its varying trend always is regarded as the most relevant variable with the change of the SC value in actual conditions [14].

Let ∆*Q*_10_, ∆*x_k_*, and ∆*ξ* be the difference of adjacent measurements on *Q*_10_, *x_k_* in KF and *ξ* from the *γ*-ray sensor, respectively. In this paper, we evaluate the consistency to ∆*Q*_10_ individually from ∆*x_k_* and ∆*ξ* by their absolute value, i.e.,
*D*(*Q*_10_, *x_k_*) = |∆*Q*_10_ − ∆*x_k_*| and *D*(*Q*_10_, *ξ*) = |∆*Q*_10_ − ∆*ξ*|(20)


According to dredging principle, an accurate estimation of SC must have as consistent value of ∆*Q*_10_ as possible. Figure 9 shows the computed values of *D*(*Q*_10_, *x_k_*) and *D*(*Q*_10_, *ξ*), respectively. It can be observed that the values of *D*(*Q*_10_, *x_k_*) are much less than those of *D*(*Q*_10_, *ξ*). Nevertheless, in about 14% sampling points, the KF values may be worse than the *γ*-ray sensor. We conclude that the following two reasons can lead to this case. Firstly, the relative 13 variables may be incomplete for general dredging conditions. In such case, the accuracy of the KF approach is surely decreased. In addition, secondly, the available measurements may be low sign-to-noise. Consequently, the convergence conditions of the KF approach is likely broken down.

With a close look, Table 4 shows the mean errors of *D*(*Q*_10_, *x_k_*) and *D*(*Q*_10_, *ξ*) at four phases of various flow patterns, and each phase contains 50 samples on average.

Compared with the *γ*-ray sensor, Table 4 shows that the overall errors of KF at the four phases has reduced with increments 0.05, 0.086, 0.078, 0.043, respectively, where the values of *D*(*Q*_10_, *x_k_*) and *D*(*Q*_10_, *ξ*) have normalized to [0, 1]. Therefore, the computed results by KF can be more acceptable in dredging engineering. The above computed results validate the KF approach.

## 5. Conclusions

In this paper, a new solid concentration (SC) estimation method for pipe flow was presented based on the KF prediction by integrating direct and indirect measurements from the *γ*-ray sensor and other relative sensors. Key factors to design a more accurate and robust CF estimation were analyzed. Results from the KF approach are closer to the real rules on a valid estimation of SC. Experiments in actual dredging engineering validate the novel method.

However, some problems remain unsolved due to incomplete flow conditions. Firstly, the general correlations from indirect measurements to SC estimation are difficult to construct and need to be analyzed further. Secondly, a real-time measurement of SC with higher accuracy is not added to the proposed method for testing, and only the rules in the sense of engineering are limited to some extent. Finally, measurements to build the proposed method only are collected from a dredging ship, and no further evaluations are performed in more working conditions. In the future, efforts will be made to solve these problems, and finding the working mechanism to calculate SC is a possible method.

## Figures and Tables

**Figure 1 sensors-20-02657-f001:**
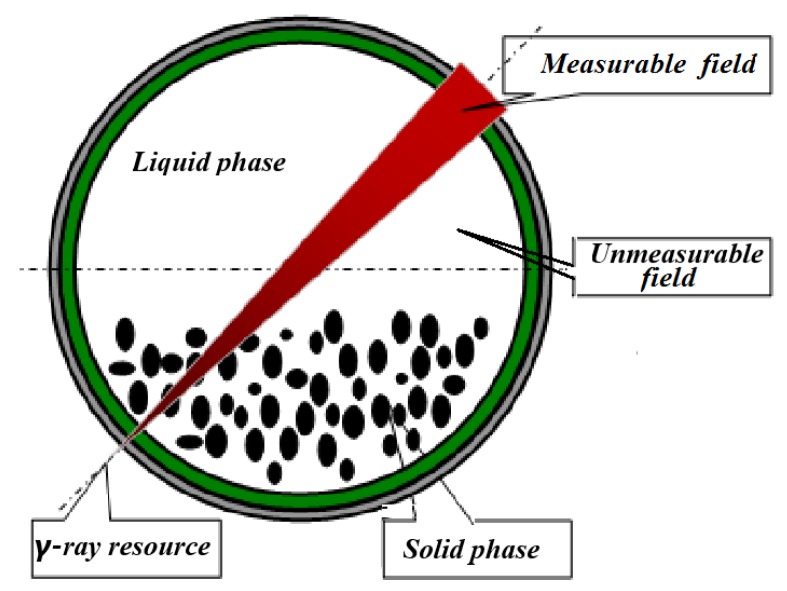
Principle of the *γ*-ray sensor.

**Figure 2 sensors-20-02657-f002:**
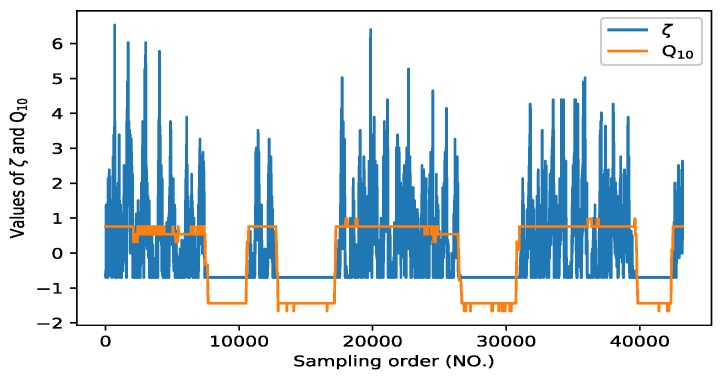
Correlation coefficient between *ξ* and *Q*_10_.

**Figure 3 sensors-20-02657-f003:**
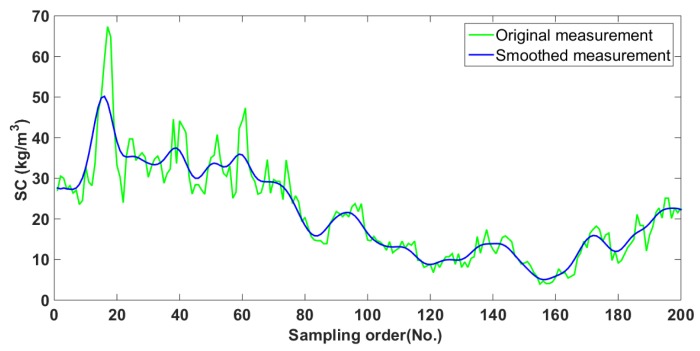
Smoothed SC measurements.

**Figure 4 sensors-20-02657-f004:**
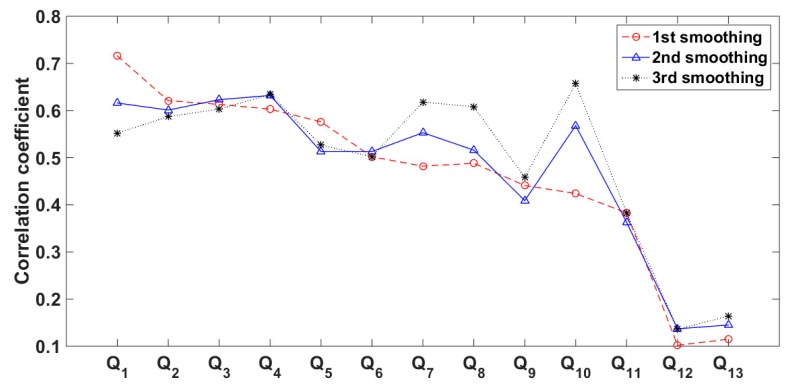
Changed correlation after smoothing measurements.

**Figure 5 sensors-20-02657-f005:**
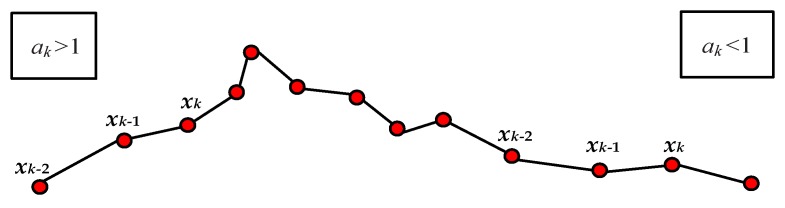
Varying trend represented by the value of *a_k__._*

**Figure 6 sensors-20-02657-f006:**
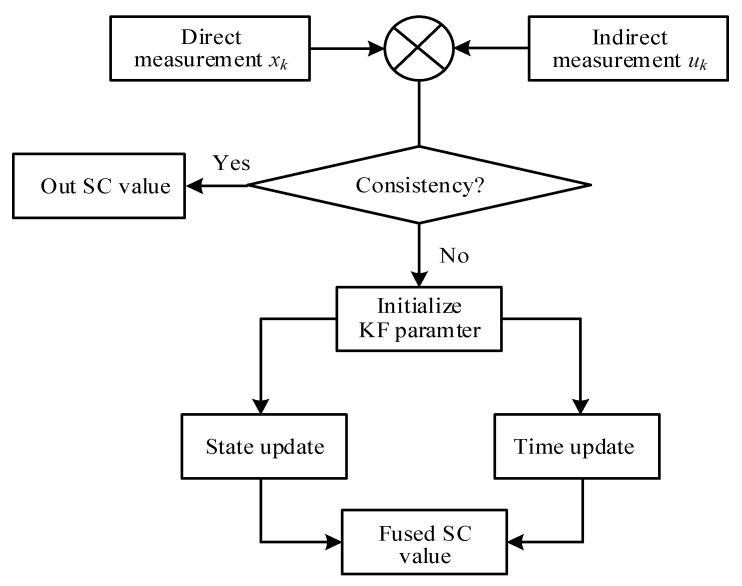
Flowchart of the KF approach.

**Figure 7 sensors-20-02657-f007:**
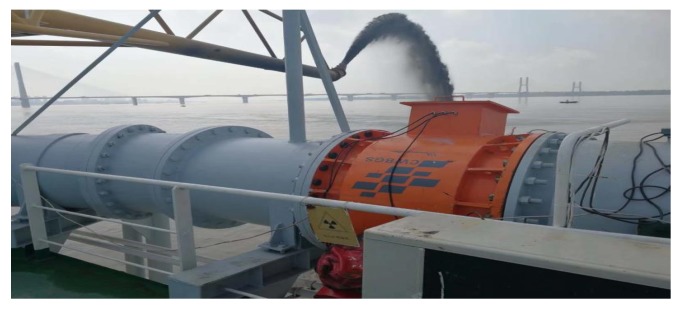
The *γ*-ray sensor in the pipeline.

**Figure 8 sensors-20-02657-f008:**
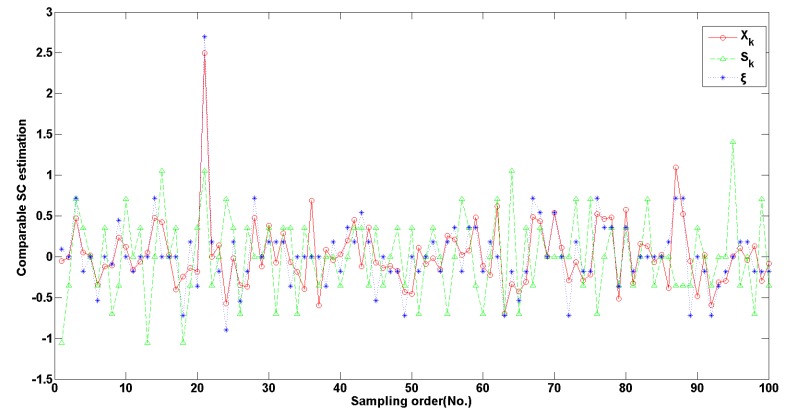
Comparison between KF for the *γ*-ray sensor.

**Figure 9 sensors-20-02657-f009:**
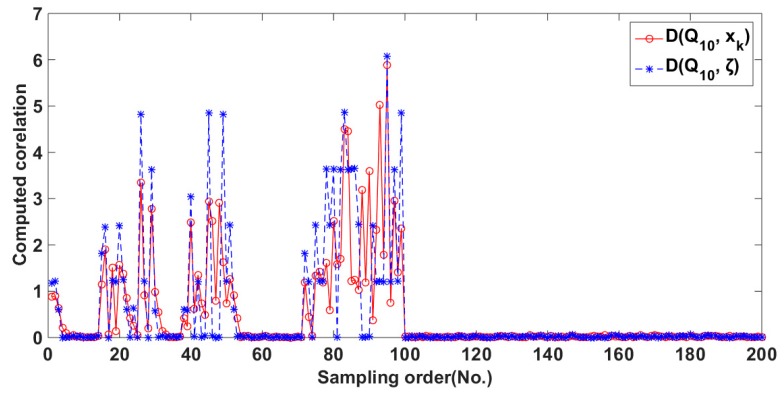
Comparison of *D*(*Q*_10_, *x_k_*) and *D*(*Q*_10_, *ξ*).

**Table 1 sensors-20-02657-t001:** Indirect measurements for solid concentration (SC) from other sensors.

Variable	Name (Unit)	*ρ* (*ξ*, *Q_k_*)
*Q* _1_	Total pressure (*k*Pa)	0.716
*Q* _2_	Reamer speed (ms^−1^)	0.621
*Q* _3_	Reamer 1 voltage (V)	0.613
*Q* _4_	Reamer 2 voltage (V)	0.603
*Q* _5_	Bridge angle (°C)	0.576
*Q* _6_	1^#^ pump speed in cabin (ms^−1^)	0.502
*Q* _7_	2^#^ pump speed in cabin (ms^−1^)	0.482
*Q* _8_	Front flushing pressure (Pa)	0.488
*Q* _9_	1^#^ mud pump power (*k*N)	0.440
*Q* _10_	Suction vacuum value(*k*Pa)	0.424
*Q* _11_	Right traverse moment (s)	0.383
*Q* _12_	Tidal level (m)	0.102
*Q* _13_	Right trunnion draught (°C)	0.115

**Table 2 sensors-20-02657-t002:** The KF prediction process.

**Input: *A_k_*, *H_k_*, *V_k_*, *W_k_*, *B_k_***
**Output: *X_k_***
1: Initialize the values of *A_k_*, *H_k_*, *V_k_*, *W_k_*;
2: Repeat
3: Compute *X_k_* and *P_k_* by Equations (6) and (7), respectively;
4: Determine *K_k_* by Equation (8);
5: Update *X_k_* by Equation (9);
6: Update *P_k_* by Equation (10).

**Table 3 sensors-20-02657-t003:** Evaluation of SC by total amount.

Number of Data	1st Day (km^3^)	2nd Day (km^3^)	3rd Day (km^3^)	4th Day (km^3^)
*Amount (x_k_*/*ξ_k_*/*s_k_)*	17.79/20.61/19.84	17.06/18.23/17.21	11.64/13.08/12.95	15.11/17.27/16.38
*Real amount*	18.63	17.56	12.23	15.79

**Table 4 sensors-20-02657-t004:** Mean errors of *D*(*Q*_10_, *x_k_*) and *D*(*Q*_10_, *ξ*) at four phases.

Amount of Data	50	100	150	200
*D*(*Q*_10_, *x_k_*)	0.781	0.908	0.664	0.503
*D*(*Q*_10_, *ξ*)	0.831	0.974	0.722	0.546

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
