# Peer review of "Solid Concentration Estimation by Kalman Filter†"

_sensors, 2020, doi:10.3390/s20092657_

Round 1
Reviewer 1 Report
- Line 31-32: why the measurement accuracy decreases when the solid and liquid distributions are not uniform? Are there any references supporting this?
- Line 43-44: what are the uncertain assumptions and conditions do the authors refer to? These should be clarified.
- Table 1: the unit of pressure should be Pa and kPa.
- Line 105: How did you calculate the covariance matrix? There are always errors in different measurements. How accurate is the matrix?
- 2: What do the X and Y axes represent? What units do they use?
- Lines 123-124: "when any measurement has sudden change, it is unbelievable". Are there any references or data supporting this? Sudden changes can be caused by many reasons, and may be due to the sudden changes in the measured variables which should be regarded as valid measurements.
- Line 140: Should it be Table 2?
- Line 143: "KF has been...". Are there any references supporting this?
- 3: What does the X axis represent?
- Lots of air bubbles can exist in the pipe in the dredging process. Are these bubbles introducing errors too? How can we reduce these errors?
- Lines 253-254: what are the other 13 sensors? What are the causes of the errors of these indirect measurements? What the accuracy of the other 13 measurement results?
- 8: What do the X and Y axes represent? What are the units? The labels are too small. This figure is very messy, and is difficult for reading and comparing.
- 9: What do the X and Y axes represent? What are the units?
- Lines 285-302: the correctness and the validity of a measurement method is evaluated not only by its stability and consistency, but also assessed by its measurement accuracy and precision. If the accuracy of the measurement is not improved, then what is the value of the proposed method? To increase the stability and consistency of the measured results, averaging methods can be beneficial too. Why using KF approach? The standards for assessing the proposed method are not totally convincing and the effectiveness of the method is thus questionable.
- Lines 306-307: Are there any literature supporting this?
- Lines 314-315, should they be Figure 9?
- Lines 332-333, The accuracy of a new measurement method is typically evaluated by a measurement method whose accuracy is much higher. Table 3 shows the difference between the two methods, but this doesn't mean that the proposed method has an improved accuracy.
- Line340: As the assessment standards and the accuracy evaluation methods are questionable, the validity of the proposed method is not completely convincing.
- Lines 341-343: what problems are unsolved? How can we address the problems in the future?
Author Response
Comments and Suggestions for Authors from Reviewer 1
- Line 31-32: why the measurement accuracy decreases when the solid and liquid distributions are not uniform? Are there any references supporting this?
Reply: Sorry for the incorrect presentation. In fact, it should is “…when the solid-liquid mixtures are not uniformly distributed”. In this situation, the averaging density in the investigated field, that is just necessarily measured, is inconsistent with the density in the line that the rays go through. Therefore, the measurements from the γ-ray sensor cannot act for the averaging density of all pixels in the entire field.
- Line 43-44: what are the uncertain assumptions and conditions do the authors refer to? These should be clarified.
Reply: Thankful for the reviewer’s careful reading. We have supplemented a typical assumption to clarify the statement in the revised mc.
- Table 1: the unit of pressure should be Pa and kPa.
Reply: They have been revised in the new mc. Thanks.
- Line 105: How did you calculate the covariance matrix? There are always errors in different measurements. How accurate is the matrix?
Reply:Sorry for the confusing. The previous statement has been rewritten by adding a calculation equation. Indeed, various flow patterns can lead to large difference of the matrixes based on measurements. But their correlations are obtained in applying all historical measurement on typical flow patterns and whereby has relatively small error. Alternatively, when the matrix has larger error, its variance in KF increases. This leads to the effect of the matrix is decreased. Thus, the relatively inaccurate matrix cannot cause large problem.
- 2: What do the X and Y axes represent? What units do they use?
Reply: Yes. These lost units and their meanings have been added.
- Lines 123-124: "when any measurement has sudden change, it is unbelievable". Are there any references or data supporting this? Sudden changes can be caused by many reasons, and may be due to the sudden changes in the measured variables which should be regarded as valid measurements.
Reply: The conclusion results from the real dredging principle. We have added the following illustrations: “ In fact, the solid-liquid mixtures have been sieved by an iron net and broken by water cannons before they are transported into the pipe. Therefore, the density change in pipe is relatively stable.” Also, all the measured variables should obey the rule. We have supplemented a reference to support this point.
- Line 140: Should it be Table 2?
Reply: Sorry. It has been corrected.
- Line 143: "KF has been...". Are there any references supporting this?
Reply: Yes. The relative references have added.
- 3: What does the X axis represent?
Reply: These lost units and their meanings have been added.
- Lots of air bubbles can exist in the pipe in the dredging process. Are these bubbles introducing errors too? How can we reduce these errors?
Reply:The measuring position of the r-ray sensor is at the beginning of the transporting pipe, where there are very large pressures. Thus, it is impossible to appear lots of bubbles.
- Lines 253-254: what are the other 13 sensors? What are the causes of the errors of these indirect measurements? What the accuracy of the other 13 measurement results?
Reply: Sorry for the previous incorrect illustrations. The sentence has been modified as “Also, the 13 indirect measurements (see Table 1) can provide an indirect estimation of SC si by Eq. (16) in real time, i= 1, 2,…,” The errors of these indirect measurements in fact results from what cite historical records in prior dataset to approximate the solution of the current 13 measurements. Also, the accuracy of the other 13 measurement results obeys the existing principle. We have further explained these in in the revised mc. Thanks.
- 8: What do the X and Y axes represent? What are the units? The labels are too small. This figure is very messy, and is difficult for reading and comparing.
Reply: These lost units and their meanings have been added.
- 9: What do the X and Y axes represent? What are the units?
Reply: These lost units and their meanings have been added.
- Lines 285-302: the correctness and the validity of a measurement method is evaluated not only by its stability and consistency, but also assessed by its measurement accuracy and precision. If the accuracy of the measurement is not improved, then what is the value of the proposed method? To increase the stability and consistency of the measured results, averaging methods can be beneficial too. Why using KF approach? The standards for assessing the proposed method are not totally convincing and the effectiveness of the method is thus questionable.
Reply:We partially agree with the reviewer’s professional insights. Indeed, the previous justifications are insufficient. We have added a new index of total product to validate our proposed method. Moreover, the previous experiment results are modified, and aim to provide clearer comparison to take on the value of our proposed method. Please see lines 306-319, in the revised mc. Thanks.
- Lines 306-307: Are there any literature supporting this?
Reply: Yes. We have added a literature to support this point.
- Lines 314-315, should they be Figure 9?
Reply: Sorry, it is been corrected. Thanks.
- Lines 332-333, The accuracy of a new measurement method is typically evaluated by a measurement method whose accuracy is much higher. Table 3 shows the difference between the two methods, but this doesn't mean that the proposed method has an improved accuracy.
Reply: Yes, the reviewer points out the keynote of the proposed method. In the traditional sense, a comparable sensor or method which has higher accuracy to test the proposed method is reasonable. But there are none such method or sensor available so far. Consequently, ones must perform an overall evaluation for the products from the sensor and other relative measurements (e.g., the 13 measurements). But these evaluation criterions fail to be fixed and nor quantifiably are computed. To evaluate the problem is just the starting point and motivations of this proposed method. In the revised mc, we have further addressed and explained these to make them clearer.
- Line340: As the assessment standards and the accuracy evaluation methods are questionable, the validity of the proposed method is not completely convincing.
Reply: Yes. Please see the reply on the comments 14 and 17. Thanks.
- Lines 341-343: what problems are unsolved? How can we address the problems in the future?
Reply: Sorry for these lost. We have supplemented these in the revised. Thanks.

Reviewer 2 Report
1. Figure 1. This figure shows the principle of the gamma-ray sensor. What is the meaning of "Unmeasurable objects"?
2. Line 89, probably it should be written SC not CS
3. Line 97: "but the seawater density is changeable in a very large range in various fields." - excluding Deas Sea with water density around 1,2 kg/dm3 sea water density is between 1,02 and 1,03 kg/dm3, so it varies around 1%. Please clarify it.
4. Figure 2 No axes descriptions,
5. Line 167: "iterative process of zk and xk are greatly affected the control action" - Not clear statement
6. Figure 3 caption. Smoothed SC measurementsnd - needs correction, remove "nd" at end
7. Line 194, doubled comma,
8. Line 231, "Then, without other prior information that their value both are unchangeable in KF. " - Not clear statement,
9. Line 235: " indirect measurements of SC are inaccurate when their value contain sudden change." - grammar error: "value contains" or "values contain"
10. Line 264: " 2) Inconsistent measurement. There are such cases that a few of measurements from the 13 sensors nearly respond to the same SC value." - I understand, that such measurements are consistent. Please clarify it.
11. Line 288: " and thus it is surely is incorrect when the estimated SC appears " - Not clear statement,
12. Figure 8. It is completely unreadable. In my opinion 100 samples is sufficient, different graphs will be more clear, axes legends are missing,
13. Line 297: "As shown in Figure 4, the fused SC" - probably Figure 8 is a correct one.
14. Lines 314, 315: " Figure 5. shows: - probably Figure 9 is a correct one.
15. Line 317: "the values of D(Q10, KF) is less than" - - grammar error: values are less
16. Line 318: "In contrast to the change trend of the SC value in Figure 8." - it is not possible to conclude it from the presented Figure 8.
17. Figure 9: as previously, no axes legend, graph will be more clear if limited to the first 100 samples.
18. Line 333: " is improved by 5%, 8.6%,7.8%,4.3%,respectively" - how these values were calculated?
19. There are no references to items 14, 17, 18 in Literature. References ale listed in a form: [1, 2] (line 24) and [12][13] (line 119), [15][16] (line 220). Please make it consistent.
There are other grammar/spelling errors, not all were commented.
Author Response
Reviewer 2 Comments and Suggestions for Authors
- Figure 1. This figure shows the principle of the gamma-ray sensor. What is the meaning of "Unmeasurable objects"?
Reply: To avoid confusion, we have revised them to “Unmeasurable field” and “measurable field”. Please see the redrawing figure in line 74 in the revised mc. Thanks.
- Line 89, probably it should be written SC not CS.
Reply: Sorry for this confusing. It has been revised.
- Line 97: "but the seawater density is changeable in a very large range in various fields." - excluding Deas Sea with water density around 1,2 kg/dm3 sea water density is between 1,02 and 1,03 kg/dm3, so it varies around 1%. Please clarify it.
Reply: Yes, we partially agree the reviewer’s suggestion; in case of relatively unchangeable conditions such as temperature, depth, etc, the changing range is small. But a dredging ship usually work from day to night under different dredging depths, it is observed that the range of density reaches to 1.01-1.08. Consequently, we think that the range is very large.
- Figure 2 No axes descriptions,
Reply: Yes. We have added the description.
- Line 167: "iterative process of zk and xk are greatly affected the control action" - Not clear statement.
Reply: Sorry for the confusion. In fact, zk is affected by the control action. We have corrected this sentence to “iterative process of xk is greatly affected by the control action whose effect is to simulate the adjustment process of direct measurement by the 13 indirect measurements. “
- Figure 3 caption. Smoothed SC measurementsnd - needs correction, remove "nd" at end
Reply: Sorry for this typing error, and it has been corrected.
- Line 194, doubled comma,
Reply: Sorry, it has been corrected.
- Line 231, "Then, without other prior information that their value both are unchangeable in KF. " - Not clear statement,
Reply: Yes, the previous sentence is unclear, and it has been corrected as “Then, without other prior information, each of them is set to an unchangeable constant in most existing KFs.” Thanks for the reviewer’s careful reading.
- Line 235: " indirect measurements of SC are inaccurate when their value contain sudden change." - grammar error: "value contains" or "values contain"
Reply: Sorry, and we have corrected it.
- Line 264: " 2) Inconsistent measurement. There are such cases that a few of measurements from the 13 sensors nearly respond to the same SC value." - I understand, that such measurements are consistent. Please clarify it.
Reply: Sorry. The previous description is incorrect. In fact, it should be “ two very different measurements of 13 sensors nearly respond to the approximately equal SC value. Thanks for the reviewer’s careful reading.
- Line 288: " and thus it is surely is incorrect when the estimated SC appears " - Not clear statement,
Reply: Sorry for the correct statement and typing error, we have supplemented the sentences to “…thus the change should be rather stable. Therefore, it is surely incorrect when the estimated SC appears local sudden change.
- Figure 8. It is completely unreadable. In my opinion 100 samples is sufficient, different graphs will be more clear, axes legends are missing,
Reply: We agree the reviewer’s suggestions, and have reduced the number of samples to 200 to show clearer figure and comparison. Moreover, some apparent signs have added to the figure.
- Line 297: "As shown in Figure 4, the fused SC" - probably Figure 8 is a correct one.
Reply: Sorry for the confusion, we have corrected it.
- Lines 314, 315: " Figure 5. shows: - probably Figure 9 is a correct one.
Reply: Yes, we have corrected it. Thanks.
- Line 317: "the values of D(Q10, KF) is less than" - - grammar error: values are less
Reply: Yes. We have corrected it. Thanks.
- Line 318: "In contrast to the change trend of the SC value in Figure 8." - it is not possible to conclude it from the resented Figure 8.
Reply: Yes. The previous conclusion is inaccurate. the change trend of the SC value is multiple levels as various the smoothing times is varied. Thus, their consistencies are multiple levels as well. Since the contrast is not key point in this subsection, we thus remove the statements.
- Figure 9: as previously, no axes legend, graph will be more clear if limited to the first 100 samples.
Reply: We agree the reviewer’s suggestions, and have reduced the number of samples to 200 to show clearer figure and comparison.
- Line 333: " is improved by 5%, 8.6%,7.8%,4.3%,respectively" - how these values were calculated?
Reply: We have formulated these presentations and whereby corrected these values, and please see lines 343- 345 in the revised mc. Thanks.
- There are no references to items 14, 17, 18 in Literature. References ale listed in a form: [1, 2] (line 24) and [12][13] (line 119), [15][16] (line 220). Please make it consistent.
There are other grammar/spelling errors, not all were commented.
Reply: All references have examined to support the relative point. Also, we have thoroughly corrected grammar and spelling in the entire mc. Thanks for the reviewer’s professional comments and suggestions.

Round 2
Reviewer 1 Report
I am pleased to see that all of my comments have been properly addressed, and recommend this paper to be accepted by the journal.